# Class-Conditional Neuron Pre-Activation Divergence to Rule out Validation Set in Label Noise Early Stopping

## Abstract

Label noise poses a major challenge in supervised deep learning: models tend to memorize corrupted labels, leading to poor generalization. Early stopping can mitigate this but usually requires a clean validation set, reducing training data. We introduce Class-Conditional Neuron Pre-Activation Divergence (CND), a metric that measures the divergence between class-conditional and marginal distributions of neuron pre-activations. We show that pre-activations naturally form class-dependent modes, which collapse during memorization of noisy labels, making the distributions less class-dependent. Leveraging this insight, we propose a validation-free early stopping criterion that relies only on training data. Specifically, we observe that the CNDs of the last layer peak near the point of maximum generalization, enabling training to be halted without a held-out validation set and thus preserving all available data for learning. Across benchmarks with symmetric and instance-dependent label noise, our method consistently outperforms other validation-free approaches—especially on datasets with many classes and at low noise levels. These results highlight the value of analyzing pre-activation distributions for understanding memorization and improving generalization.

## 1 Introduction

Label noise is a common challenge in supervised classification, eroding the generalization ability of deep learning models. Among mitigation strategies, early stopping remains especially effective because it is simple, does not require major changes to the training algorithm, and pairs easily with other methods. Deep Neural Network (DNN)s follow distinct learning dynamics when trained on datasets in which some samples contain label noise: they first capture simple, generalizable patterns from well-labeled data while largely ignoring mislabeled examples, then gradually begin fitting the noisy labels, which leads to overfitting and degraded generalization Li et al. (2020b). Early stopping naturally counteracts this memorization effect. Its main limitation is its reliance on a noise-free validation set, which requires withholding clean training data. Larger validation sets provide more reliable performance estimates, but they further reduce the data available for learning. Toner & Storkey (2024) show that, under typical noise models, accuracy on a noisy validation set can stand in for a clean one, albeit still at the cost of holding out valuable training data.

Following this motivation, Yuan et al. (2023a) recently introduced the first early stopping technique designed to mitigate the detrimental effects of label noise without requiring a separate validation dataset. This method is based on the observation that the peak in generalization often coincides with an increase in the instability of the network's predictions, measured by a proposed metric called Prediction Changes (PC), which can be monitored directly on the training set. However, this approach may fail to identify the generalization peak in cases where the dataset contains little label noise.

Label noise is often used to probe memorization in deep learning because neural networks can readily memorize an entire training set, and fitting correctly labeled examples in noisy data requires memorizing those instances. Its presence therefore provides a unique opportunity to delineate the boundary between generalization and memorization (Maini et al., 2023; Baldock et al., 2021; Stephenson et al., 2021). Numerous studies have examined memorization in DNNs, focusing on how networks learn from corrupted labels and where such samples are stored within the network

(Arpit et al., 2017; Zhang et al., 2021; Feldman & Zhang, 2020; Stephenson et al., 2021; Baldock et al., 2021). Notably, Stephenson et al. (2021), Baldock et al. (2021), and Feldman & Zhang (2020) found that generalizable samples tend to be represented in early layers, closer to the input, while harder-to-learn or noisy examples—typically memorized ones—are represented in deeper layers, nearer to the output. In contrast, Maini et al. (2023) reported that memorization is localized in a small set of intermediate neurons. Understanding how DNNs transition between generalization and memorization thus offers valuable insights into their learning dynamics and may pave the way toward more robust models. In this context, analyzing neuron pre-activation distributions has proven to be a powerful approach for distinguishing meaningful generalization from spurious or memorized patterns, as deviations in these distributions can expose abnormal neurons (Zheng et al., 2022) and their coverage strongly correlates with a model's generalization ability (Liu et al., 2024).

Inspired by previous work, we study the relation between neuron pre-activation distributions and label-noise memorization in multiclass supervised learning. We ask whether treating pre-activations as random variables reveals class-dependent structure in their distributions, and hypothesize that this dependency weakens as memorization sets in. We provide evidence that pre-activation distributions are better modeled as class-dependent mixtures: when the model generalizes, internal representations display strong class specificity, but as memorization progresses, they shift toward less class-specific patterns. To capture this transition, we introduce a novel metric, the Class-Conditional Neuron Pre-Activation Divergence (CND), which quantifies divergence between class-conditional pre-activation distributions. It is a local neuron-level metric based on the Jensen-Shannon Divergence (JSD). We analyze its evolution across training epochs and layers, and leverage it to propose a validation-free early stopping criterion that preserves clean samples, based on the observation that CNDs in the last hidden layer peak in alignment with generalization. We compare our method with PC and with another technique introduced here, Known Polluted Accuracy (KPA), which augments training with a small subset of deliberately mislabeled examples whose true labels are known, allowing us to track their accuracy. Our experiments reveal that CND outperforms the other solutions, particularly on datasets with many classes, and exhibits greater robustness to hyperparameter choices.

The main contributions of this work are summarized as follows:

1. We provide a new characterization of neuron pre-activation distributions, showing that in multiclass classification they can be modeled as mixtures, where each mode corresponds to a different class.

2. We introduce the Class-Conditional Neuron Pre-Activation Divergence (CND), a local, per-neuron metric based on the JSD that quantifies divergence between class-conditional pre-activation distributions and reveals how memorization weakens class-specific structure in hidden representations.

3. We propose a validation-free early stopping criterion based on the peak CND in the last hidden layer, which preserves all clean training data and outperforms existing approaches, especially on datasets with many classes.

## 2 RELATED WORK

Strategies for mitigating label noise can be broadly grouped into five categories: robust loss functions, such as Mean Absolute Error (MAE), Symmetric Cross Entropy (SCE), and Generalized Cross Entropy (GCE), which reduce sensitivity to corrupted labels (Ghosh et al., 2017; Wang et al., 2019); sample selection methods, including Co-Teaching and MentorNet, which prioritize clean examples during training (Han et al., 2018; Jiang et al., 2018); noise adaptation layers, which explicitly model label corruption (Goldberger & Ben-Reuven, 2017); semi-supervised approaches, such as DivideMix, which separate clean and noisy samples using mixture models (Li et al., 2020a); and regularization techniques and early stopping, where methods like dropout, data augmentation, or halting training early help prevent memorization of noisy labels (Srivastava et al., 2014; Arpit et al., 2017; Li et al., 2020b; Yuan et al., 2023b). In this work, we focus specifically on validation-free early stopping techniques.

A recent approach, Label Wave (Yuan et al., 2023a), introduced an early stopping criterion based on prediction stability, without requiring a validation set. The key metric, *Prediction Change*, is defined

as

$$PC_t = \sum_{i \in \mathcal{D}} \left( 1 - \delta \left( \hat{y}_i^{(t)}, \hat{y}_i^{(t-1)} \right) \right), \tag{1}$$

where $\mathcal{D}$ denotes the training set, $\hat{y}_i^{(t)}$ is the predicted label for sample $i$ at epoch $t$, and $\delta(a, b)$ is the Kronecker delta function, equal to 1 if $a = b$ and 0 otherwise. Thus $PC_t$ counts the number of training samples whose predicted label changes between consecutive epochs. A local minimum of $PC_t$ often signals the onset of memorization. To improve reliability, the authors propose a smoothed version with a moving average and patience.

Beyond standard early stopping, Bai et al. (2021) showed that later layers are more susceptible to label noise than earlier ones, and proposed Progressive Early Stopping (PES), which progressively freezes and retrains layers with decreasing epochs.

The phenomenon of memorization in DNNs has been analyzed from several perspectives. Feldman & Zhang (2020) quantified memorization at the sample level, while Arpit et al. (2017) leveraged label noise to study memorization more broadly. Layer-wise analyses have shown that generalization emerges in early layers, whereas later layers tend to memorize harder or noisier examples (Stephenson et al., 2021; Baldock et al., 2021). The *prediction depth* metric (Baldock et al., 2021) formalizes this by identifying the layer at which a sample's classification stabilizes. Other work has localized memorization to specific intermediate neurons, identified through adversarial-style perturbations (Maini et al., 2023), with important implications for privacy, fairness, and robustness.

Finally, the study of neuron pre-activation distributions connects theory and empirical practice. Neural networks propagate information through layers using weighted sums of neuron activations followed by non-linear transformations. With *pre-activations*, we refer to the intermediate values before applying non-linearities. In a fully connected neural network, each neuron in layer $l$ receives inputs from all neurons in the previous layer $l - 1$, producing a pre-activation computed as:

$$z_j^{(l)} = \sum_{i=0}^{n_{l-1}} w_{ij}^{(l)} a_i^{(l-1)}, \tag{2}$$

In convolutional layers, we refer to pre-activation of neuron $\bar{z}_k^{(l)}$ as the average pooling of the filter $k$ in layer $l$ :

$$\bar{z}_k^{(l)} = \frac{1}{P \cdot P} \sum_{m=1}^{P} \sum_{n=1}^{P} z_{k,m,n}^{(l)}, \tag{3}$$

with $z_{k,m,n}^{(l)}$ is the pre-activation at spatial position $(m, n)$ for filter $k$ in layer $l$. Gaussianity of these pre-activation distributions is a common assumption in analyses such as Neural Tangent Kernel (NTK) and the Edge of Chaos (Jacot et al., 2018; Poole et al., 2016), but empirical evidence shows heavier-tailed distributions in finite-width networks (Wolinski & Arbel, 2025; Vladimirova et al., 2019). Enforcing Gaussian priors can even harm performance, while Laplace-like distributions improve robustness (Fortuin et al., 2022). A first practical application arises in backdoor detection: neurons compromised by triggers show bi-modal pre-activation distributions, making them identifiable and suppressible (Zheng et al., 2022). Another application is out-of-distribution (OOD) detection. Liu et al. (2024) proposed Neuron Activation Coverage (NAC) which measures how well test-time neuron activations are covered by those induced from in-distribution training data. OOD inputs typically fall outside this coverage, yielding lower scores. Moreover, NAC correlates positively with model generalization ability, further highlighting the potential of analyzing neuron pre-activation distributions.

## 3 CLASS-CONDITIONAL PRE-ACTIVATION DISTRIBUTIONS

In this work, we aim to investigate how the pre-activation random variable is conditioned on the ground truth class and how this information can be used as a proxy for memorization. The key idea is that the pre-activation $z_j^{(l)}$ of a neuron $j$ in hidden layer $l$ follows a class-conditional distribution $p(z_j^{(l)} \mid y)$, which captures meaningful patterns associated with the true class $y$. As the model begins to memorize noisy training data, this conditional distribution gradually weakens its dependency on $y$

and shifts toward the marginal distribution $p(z_j^{(l)})$. This transition indicates that neuron activations become increasingly sensitive to spurious input patterns rather than robust class-discriminative features. Ultimately, this shift leads to a degradation in the model's ability to learn meaningful representations and increases the risk of overfitting.

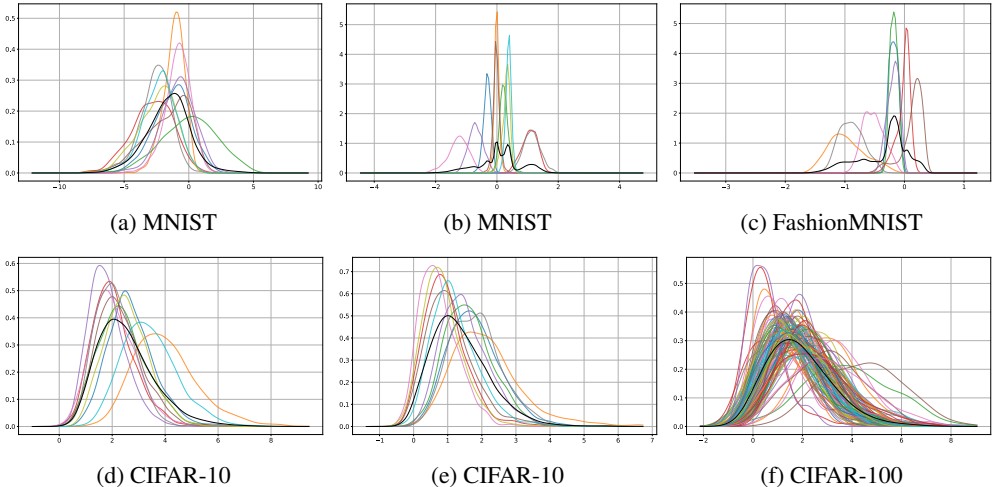

(a) MNIST        (b) MNIST       (c) FashionMNIST

(d) CIFAR-10      (e) CIFAR-10      (f) CIFAR-100

Figure 1: Class-conditional pre-activation distributions for representative neurons across datasets. Each colored curve is a class-specific pre-activation Probability Density Function (PDF); the number of curves matches the number of classes, and multimodality indicates class separation. The x-axis reports the neuron pre-activation value, and the y-axis denotes the estimated density. Networks and layers per panel: MNIST MLP (a) first layer, (b) last hidden layer; FashionMNIST MLP (c) last hidden layer; CIFAR-10 ResNet9 (d) conv feature map (pre-activation of one filter), (e) last hidden layer; CIFAR-100 ResNet9 (f) last hidden layer.

To test our hypothesis that neuron pre-activation distributions are class-dependent and capture class-specific patterns, we analyze these distributions across layers, datasets, and network architectures. As shown in Figure 1, the results confirm that pre-activation distributions preserve class-specific structure. Clear separation between classes emerges even at the level of individual neurons, rather than only when activations are aggregated across layers, as emphasized in prior work (Stephenson et al., 2021; Baldock et al., 2021). This provides a more fine-grained interpretation of neuron activation PDF, compared to previous studies that focused primarily on class-marginal distributions. Complementary statistical analyses reported in Appendix A further validate these conclusions.

Our findings are also aligned with the observations of Zheng et al. (2022), who showed that neurons with bimodal pre-activation distributions are responsible for memorizing backdoor attacks. Our results are consistent with this perspective, since a backdoor trigger can be interpreted as introducing an additional "pseudo-class" that yields a distinct activation distribution mode.

## 4 CLASS-CONDITIONAL NEURON PRE-ACTIVATION DIVERGENCE

We now analyze how neuron pre-activation distributions evolve as a network begins to memorize label noise. Specifically, we track the distributions of a hidden neuron at two critical training stages: the point of maximum generalization and the final epoch, when the model has memorized the noisy labels. Early in training, DNNs tend to fit clean samples first, which drives a surge in accuracy; as training proceeds, they start to memorize mislabeled points, causing generalization to drop. Figure 2 shows representative class-conditional distributions at these two stages. When the model generalizes well, the divergence between class-conditional distributions is maximized. During memorization, this distance shrinks and the distributions collapse toward the marginal distribution, even though their supports remain broad. This analysis spans multiple datasets, training strategies, network architectures, and hidden neurons.

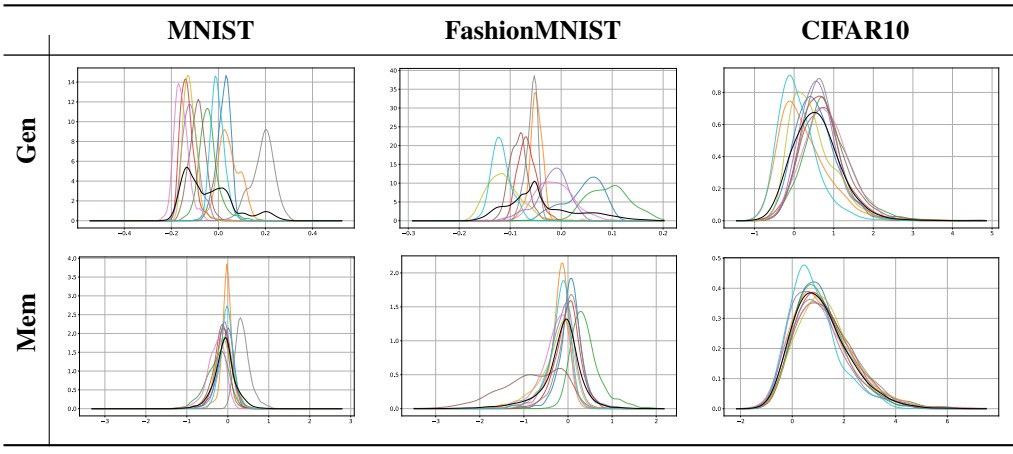

Figure 2: Class-conditional pre-activation distributions of a representative hidden neuron at maximum generalization (Gen) and after memorizing noisy labels (Mem). Each color is a class; divergence is high at Gen and shrinks toward the marginal at Mem.

Based on these intuitions, we introduce a metric to quantify this phenomenon by measuring the distance between class-conditional and marginal pre-activation distributions. We call this metric the Class-Conditional Neuron Pre-Activation Divergence (CND). Formally, for a given layer $l$ and neuron $j$, CND is defined as the generalized JSD:

$$\mathrm{CND}_l[j](\mathcal{B}) = \mathrm{JSD}_{\boldsymbol{\pi}}\Big(\{P(z_j^{(l)} \mid y)\}_{y \in \mathcal{Y}}\Big) \tag{4}$$

$$= \sum_{y \in \mathcal{Y}} \pi_y \, D_{\mathrm{KL}}\Big(P(z_j^{(l)} \mid y) \,\Big\|\, P(z_j^{(l)})\Big), \tag{5}$$

where $\mathcal{B}$ denotes the batch used for estimation and $\boldsymbol{\pi} = (\pi_y)_{y \in \mathcal{Y}}$ represents the class weights, matching the class prior (for balanced datasets, $\boldsymbol{\pi}$ is uniform). The associated mixture distribution is $M_j^{(l)}(z) = \sum_{y \in \mathcal{Y}} \pi_y \, P(z_j^{(l)} \mid y)$, which matches the true marginal $P(z_j^{(l)})$ if and only if the empirical class frequency equals $\pi_y$ for all $y$.

We deliberately compute CND on the full training set—including mislabeled instances—to avoid reliance on a clean subset. Consequently, the observed conditionals are $P(z_j^{(l)} \mid \tilde{y})$, where $\tilde{y}$ is the observed (possibly corrupted) label. The JSD is a symmetrized and smoothed version of the Kullback-Leibler Divergence (KL) divergence. It is always finite and bounded between $0$ and $\mathbb{H}[\boldsymbol{\pi}]$ (the entropy of the weights), attaining $0$ if and only if all class-conditional distributions coincide. Importantly, unlike KL, the JSD naturally extends to comparing more than two distributions, which is crucial in our setting where multiple class-conditional densities must be contrasted with the marginal.

The value of $\mathrm{CND}_l[j](\mathcal{B})$ quantifies how distinct the class-conditional distribution of a neuron's activations is from the marginal distribution. Higher values indicate neurons that exhibit stronger class separability in their activation patterns. To analyze how CND evolves during training, Figure 3 presents its dynamics across multiple datasets and architectures, grouping neurons by layer and considering both clean- and noisy-label settings. We find that the CNDs of the final hidden layer most closely track the generalization behavior of the DNN. Alongside the training curves, we show the evolution of CND for each layer and report layer-wise averages computed as the mean across neurons, with shaded regions indicating one standard deviation. This visualization highlights both the internal variability within layers and their relationship to generalization performance. Across all experiments, the final layer provides the clearest signal of generalization. In the absence of label noise (top row), final-layer CND values remain stable. In contrast, under label noise (middle and bottom rows), peaks in the final-layer CND coincide with the point of maximal generalization and then decline in close alignment with decreases in test accuracy. This behavior is consistent with Bai et al. (2021), who showed that later layers are more susceptible to label noise.

Two key observations emerge: first, CND values typically increase from early to deeper layers, suggesting that deeper neurons encode more class-discriminative information; second, this makes the final layer's CND a strong candidate for validation-free early stopping based solely on training data. Studying neuron PDFs therefore reveals functional roles tied to generalization versus memorization. Moreover, neuron pruning experiments (Appendix G), where neurons are removed based on their CND value, further show that removing high-CND neurons sharply degrades generalization, whereas memorization is more diffuse: pruning low-CND neurons primarily harms clean training accuracy while leaving test accuracy relatively stable.

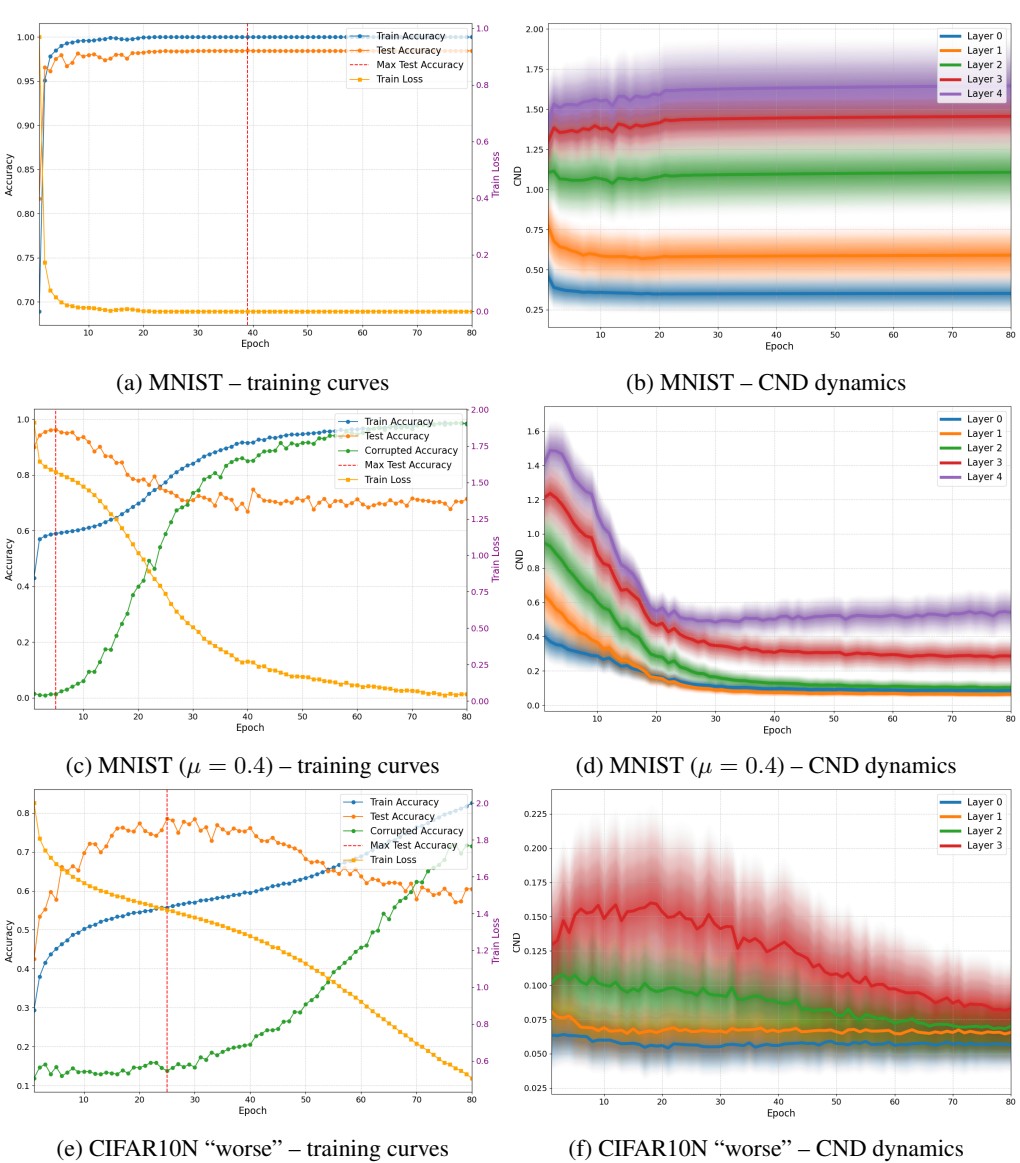

(a) MNIST – training curves

(b) MNIST – CND dynamics

(c) MNIST ($\mu = 0.4$) – training curves

(d) MNIST ($\mu = 0.4$) – CND dynamics

(e) CIFAR10N "worse" – training curves

(f) CIFAR10N "worse" – CND dynamics

Figure 3: Training curves (left) and CND dynamics (right) for different experiments. Top row: a fully connected DNN trained on MNIST. Middle row: the same architecture trained on MNIST with symmetric label noise at rate $\mu = 0.4$. Bottom row: a ResNet9 trained on CIFAR10N with instance-dependent label noise ("worse" labels). "Corrupted accuracy" denotes the accuracy measured only on samples affected by label noise, for the KPA metrics introduced in Section 5.

## 5 EXPERIMENTS

Building on the previous observations, this section shows how the CND metric can be used as an early stopping criterion and contrasts it with alternatives that rely solely on training-set information. In all cases, these approaches eliminate the need for a validation set, thereby preserving valuable training data.

In addition to the CND, we propose an alternative early stopping metric: the KPA. This metric, used alongside PC and CND, relies on a subset of deliberately polluted labels that are duplicated and reintroduced into the dataset. Specifically, let $\mathcal{S}_{\text{clean}} = \{(\mathbf{x}_i, y_i)\}_{i=1}^N$ denote the original training dataset, where $y_i$ is the true label for input $\mathbf{x}_i$. We define a polluted subset $\mathcal{S}_{\text{polluted}} \subset \{(\mathbf{x}_i, \tilde{y}_i)\}$, where each $\tilde{y}_i \neq y_i$, created by selecting a subset of the original samples and assigning them incorrect random noisy labels. These polluted samples are duplicates of original ones with altered labels, not replacements. The final training set becomes the union $\mathcal{S}_{\text{train}} = \mathcal{S}_{\text{clean}} \cup \mathcal{S}_{\text{polluted}}$, ensuring that all original data is preserved while introducing controlled noise for monitoring purposes.

The KPA is then defined as

$$\text{KPA}(t) = \frac{1}{|\mathcal{S}_{\text{polluted}}|} \sum_{(\mathbf{x}_i, \tilde{y}_i) \in \mathcal{S}_{\text{polluted}}} \delta\left(f_t(\mathbf{x}_i), \tilde{y}_i\right). \tag{6}$$

Here $f_t(\cdot)$ is the model's prediction function at training epoch $t$. An increase in $\text{KPA}(t)$ over time signals that the model is beginning to memorize noisy labels, which is an indicator to consider early stopping.

Early stopping prevents overfitting by halting training when a monitored metric begins to deteriorate. To enhance stability across all three metrics, we apply a moving average smoothing technique. Based on the metric used, training is stopped at the first local minimum (KPA and PC) or the first local maximum (CND of the last hidden layer). The pseudo-algorithm is shown in Appendix B.

We evaluate CND-based early-stopping criteria on a range of noisy-label benchmarks. For image classification, we use CIFAR-10 and CIFAR-100 with synthetic symmetric label noise, their real, instance-dependent counterparts CIFAR-10N and CIFAR-100N, as well as Tiny-ImageNet (synthetic symmetric noise) and Mini-WebVision (naturally noisy web-scale image classification). To demonstrate that our analysis is not limited to image data, we also conduct experiments on the 20 Newsgroups (NEWS20) text classification dataset with synthetic symmetric noise. CIFAR experiments use a Preact-ResNet18, Tiny-ImageNet and Mini-WebVision both use a ResNet-50 backbone, and NEWS20 uses a 3-layer MLP. Dataset details and simulation settings are provided in Appendices C and D. Unless otherwise stated, we focus on symmetric noise levels up to $40\%$, matching the worst corruption observed in the real, instance-dependent CIFAR-10N/100N annotations and thus representing a realistic operating regime. Beyond that range, the CND signal measured on the noisy training set loses contrast because layer-wise conditionals become dominated by mislabeled samples. Appendix F shows that CND can also be successfully applied under extremely high noise levels by recomputing CND only on a small clean subset injected into training. Although this approach still requires access to a known clean subset of the data, it does not reserve that subset as a held-out validation set, but instead uses all available data during training.

### 5.1 RESULTS AND OBSERVATIONS

Results are reported in Tables 1a, 1b, and 1c, where $e$ denotes the selected epoch, $\Delta e$ its signed distance from the epoch of peak test accuracy, $a$ the test accuracy at $e$, and $\Delta a$ the accuracy drop relative to the best-performing model. Negative $\Delta e$ indicates premature stopping, while positive values indicate late stopping. All results are averaged over multiple runs.

On CIFAR-10 with symmetric noise, CND and PC perform similarly at low to moderate noise rates. At $10\%$ and $20\%$, CND selects epochs closer to the optimum and achieves slightly higher accuracy, while PC tends to stop later with higher variance. At $40\%$, the two are comparable. Under instance-dependent noise, PC performs better in the "agree" case, but here the optimal epoch does not align with memorization onset, which CND captures. Overall, CND is closer to the optimal epoch in most settings, while PC often overshoots and KPA consistently stops too early, leading to the largest accuracy losses. Results for extreme synthetic noise rates obtained via the clean-subset protocol are deferred to Appendix F.

On CIFAR-100, PC generally fails to identify the optimum, usually stopping far too late—except at 10% symmetric noise, where it performs best. CND, by contrast, consistently selects more appropriate epochs at moderate and high noise levels, while KPA again stops prematurely. As shown in Figure 6a, CND can detect memorization onset even when this does not coincide with maximum generalization.

On NEWS, the three methods are more balanced, with CND slightly ahead in accuracy. No clear winner emerges between CND and PC, but these results confirm that CND is effective beyond image data, extending to text classification as well.

Finally, we note differences in how the metrics operate. PC is highly sensitive to its patience parameter: if set too high, it may miss early signals of overfitting. CND instead requires an aggregation rule for neuron-wise values in the last layer; we report high-quantile (upper quartile) summaries because Appendix G shows that neurons with the largest CND values are the ones most responsible for downstream generalization, so emphasizing their trajectories enhances sensitivity to memorization. Exploring alternative aggregation strategies and richer uses of pre-activation distributions is an interesting avenue for future work.

To further stress-test the metrics on web-scale data, we also apply the same early-stopping procedure to Mini-WebVision, with results summarized in Table 2. As observed, PC does not identify the point of maximum generalization, whereas CND maintains training closer to that peak and achieves accuracies nearest to the optimal stopping point. In contrast, KPA triggers too early, leading to premature stopping and noticeably degraded performance. We additionally evaluate the metrics on Tiny-ImageNet under $10\% - 40\%$ symmetric label noise, with results summarized in Table 2. At 10% label noise, the maximum generalization performance does not coincide with the first peak because the noise level is relatively mild; instead, test accuracy continues to increase later in training and reaches a second, higher maximum. In this regime, CND stops training too early and therefore yields suboptimal generalization. For higher noise levels, CND holds training closer to the point of maximum generalization than the other methods, achieving the highest generalization accuracy at $30\% - 40\%$ noise. PC always misses the memorization onset, holding training almost until the end, when the metric has already stabilized near a low local minimum.

In conclusion, the CND metric is generally more stable and effective at detecting the peak of generalization performance; however, it fails to capture secondary increases in accuracy that may occur after the initial drop caused by memorization. The PC metric is highly sensitive to hyperparameter choices and performs poorly under severe label noise. In contrast, the KPA metric provides early detection of memorization onset but often halts training prematurely. Understanding these trade-offs is crucial for selecting an appropriate early stopping strategy tailored to the noise characteristics of a given dataset.

To illustrate the behavior of the proposed early stopping metrics, we include a subset of representative training curves in the main body (full results are provided in Appendix E). Figure 4 shows results on CIFAR-10 with two instance-dependent noise settings (*Rand 1* and *Worse*) and on CIFAR-100 with symmetric noise at 40% and with *Worse* labels. Across all cases, the CND curve in the last hidden layer peaks in close alignment with maximum test accuracy, whereas PC often exhibits a plateau instead of a local minima, which makes fail the PC detection of maximum accuracy. Meanwhile KPA tends to stop prematurely.

## 6 CONCLUSIONS

We introduced CND, a metric based on neuron pre-activation distributions, and demonstrated its effectiveness as a validation-free early stopping criterion. Across benchmarks with both symmetric and instance-dependent label noise, CND consistently identified epochs closer to optimal generalization than existing methods, particularly on datasets with many classes. Beyond early stopping, CND also provided insights into memorization dynamics by highlighting neurons that primarily fit noisy labels. Our study has some limitations: the choice of aggregation strategy for neuron-wise values may influence performance; CND does not capture the secondary increase in accuracy that sometimes occurs late in training; and under very high (arguably unrealistic) label noise, the CND signal on the training data fails to offer a reliable stopping criterion.

Table 1: Early Stopping Results for CIFAR-10, CIFAR-100, and NEWS.

| noise type | label noise ratio | metric | e
mean ± std | $\Delta e$
mean ± std | a
mean ± std | $\Delta a$
mean ± std |
|---|---|---|---|---|---|---|
| symmetric | 10% | KPA | 22.00 ± 3.74 | -14.20 ± 7.33 | 76.09% ± 7.20% | 9.25% ± 6.06% |
| | | PC | 42.80 ± 9.09 | 6.60 ± 11.22 | 82.65% ± 3.12% | 2.69% ± 1.85% |
| | | CND | 35.20 ± 5.81 | -1.00 ± 3.32 | 83.26% ± 2.02% | 2.08% ± 2.00% |
| | 20% | KPA | 24.60 ± 6.43 | -13.60 ± 7.80 | 80.29% ± 3.50% | 3.50% ± 1.59% |
| | | PC | 43.20 ± 1.92 | 5.00 ± 5.24 | 82.15% ± 2.09% | 1.63% ± 1.08% |
| | | CND | 37.00 ± 5.05 | -1.20 ± 4.55 | 82.67% ± 2.82% | 1.12% ± 0.93% |
| | 30% | KPA | 15.40 ± 3.51 | -9.20 ± 4.60 | 82.28% ± 0.90% | 2.49% ± 0.86% |
| | | PC | 14.20 ± 4.66 | -10.40 ± 6.54 | 82.44% ± 1.74% | 2.33% ± 1.81% |
| | | CND | 16.80 ± 1.48 | -7.80 ± 3.03 | 83.47% ± 0.65% | 1.30% ± 0.64% |
| | 40% | KPA | 24.60 ± 6.35 | -7.80 ± 7.89 | 70.47% ± 2.80% | 5.02% ± 3.52% |
| | | PC | 36.80 ± 7.05 | 4.40 ± 7.54 | 74.83% ± 1.64% | 0.67% ± 0.60% |
| | | CND | 38.40 ± 5.59 | 6.00 ± 6.20 | 73.79% ± 2.24% | 1.70% ± 1.07% |
| aggre label | 9.01% | KPA | 23.20 ± 4.55 | -22.20 ± 25.08 | 80.05% ± 5.02% | 5.68% ± 4.48% |
| | | PC | 86.60 ± 17.87 | 41.20 ± 24.73 | 83.66% ± 2.31% | 2.07% ± 1.96% |
| | | CND | 38.40 ± 4.28 | -7.00 ± 23.93 | 82.55% ± 1.91% | 3.18% ± 1.75% |
| random label1 | 17.23% | KPA | 22.40 ± 5.03 | -11.00 ± 1.87 | 81.36% ± 1.14% | 2.41% ± 0.88% |
| | | PC | 57.00 ± 23.72 | 23.60 ± 23.23 | 80.80% ± 2.23% | 2.97% ± 1.44% |
| | | CND | 30.00 ± 3.16 | -3.40 ± 4.98 | 82.78% ± 2.22% | 0.98% ± 1.33% |
| random label2 | 18.12% | KPA | 20.20 ± 9.09 | -14.40 ± 6.73 | 77.17% ± 4.49% | 5.44% ± 3.96% |
| | | PC | 43.60 ± 4.16 | 9.00 ± 5.83 | 81.24% ± 2.08% | 1.37% ± 1.08% |
| | | CND | 30.40 ± 3.36 | -4.20 ± 4.87 | 81.23% ± 1.75% | 1.38% ± 0.87% |
| random label3 | 17.64% | KPA | 19.80 ± 5.85 | -14.40 ± 5.98 | 80.44% ± 1.41% | 3.49% ± 2.27% |
| | | PC | 52.00 ± 26.65 | 17.80 ± 28.17 | 81.75% ± 2.30% | 2.18% ± 1.13% |
| | | CND | 29.80 ± 3.42 | -4.40 ± 5.41 | 82.19% ± 1.82% | 1.74% ± 1.30% |
| worse label | 40.21% | KPA | 18.60 ± 6.58 | -10.60 ± 10.81 | 70.24% ± 4.83% | 6.35% ± 4.08% |
| | | PC | 30.60 ± 4.34 | 1.40 ± 3.21 | 75.27% ± 1.99% | 1.31% ± 1.04% |
| | | CND | 27.00 ± 5.57 | -2.20 ± 7.50 | 74.59% ± 0.97% | 1.99% ± 0.82% |

(a) CIFAR-10

| noise type | label noise ratio | metric | e
mean ± std | $\Delta e$
mean ± std | a
mean ± std | $\Delta a$
mean ± std |
|---|---|---|---|---|---|---|
| label noise | 10% | KPA | 10.00 ± 3.39 | -82.60 ± 4.77 | 43.52% ± 6.96% | 16.92% ± 7.59% |
| | | PC | 91.00 ± 9.06 | -1.60 ± 8.71 | 59.72% ± 1.04% | 0.72% ± 0.79% |
| | | CND | 14.80 ± 2.28 | -77.80 ± 5.31 | 53.04% ± 2.05% | 7.40% ± 1.88% |
| | 20% | KPA | 8.20 ± 3.11 | -9.20 ± 4.02 | 45.86% ± 7.47% | 11.13% ± 7.24% |
| | | PC | 98.00 ± 1.00 | 80.60 ± 1.67 | 53.51% ± 0.63% | 3.48% ± 0.57% |
| | | CND | 14.20 ± 2.17 | -3.20 ± 2.17 | 54.57% ± 1.58% | 2.42% ± 1.49% |
| | 30% | KPA | 6.80 ± 2.68 | -10.00 ± 3.74 | 38.71% ± 6.60% | 14.65% ± 6.69% |
| | | PC | 98.20 ± 1.30 | 81.40 ± 2.30 | 46.40% ± 0.40% | 6.96% ± 0.62% |
| | | CND | 11.00 ± 3.74 | -5.80 ± 4.76 | 47.15% ± 6.35% | 6.21% ± 6.14% |
| | 40% | KPA | 12.80 ± 4.49 | -6.60 ± 4.51 | 43.08% ± 6.06% | 5.27% ± 5.65% |
| | | PC | 83.20 ± 34.22 | 63.80 ± 33.45 | 39.82% ± 4.27% | 8.53% ± 3.81% |
| | | CND | 15.40 ± 1.82 | -4.00 ± 1.22 | 45.63% ± 1.30% | 2.72% ± 1.55% |
| worse label | 40.2% | KPA | 8.40 ± 4.28 | -9.20 ± 3.77 | 40.74% ± 6.34% | 10.15% ± 6.10% |
| | | PC | 98.40 ± 0.89 | 80.80 ± 2.77 | 46.22% ± 0.59% | 4.67% ± 0.97% |
| | | CND | 16.60 ± 1.14 | -1.00 ± 2.55 | 49.84% ± 0.70% | 1.05% ± 0.83% |

(b) CIFAR-100

| noise type | label noise ratio | metric | e
mean ± std | $\Delta e$
mean ± std | a
mean ± std | $\Delta a$
mean ± std |
|---|---|---|---|---|---|---|
| label noise | 20% | KPA | 10.60 ± 5.32 | -14.20 ± 8.35 | 75.91% ± 1.56% | 2.10% ± 1.66% |
| | | PC | 16.20 ± 6.18 | -8.60 ± 10.01 | 76.63% ± 0.89% | 1.37% ± 1.13% |
| | | CND | 15.00 ± 5.43 | -9.80 ± 7.19 | 77.01% ± 0.29% | 0.99% ± 0.43% |
| | 30% | KPA | 12.00 ± 5.24 | -11.20 ± 8.32 | 75.04% ± 1.06% | 1.75% ± 1.16% |
| | | PC | 13.80 ± 7.46 | -9.40 ± 12.22 | 74.55% ± 0.53% | 2.23% ± 0.72% |
| | | CND | 20.20 ± 9.31 | -3.00 ± 13.78 | 75.18% ± 0.93% | 1.61% ± 1.04% |
| | 39.99% | KPA | 16.20 ± 10.18 | -3.40 ± 11.33 | 74.16% ± 0.34% | 0.93% ± 0.53% |
| | | PC | 16.00 ± 6.75 | -3.60 ± 9.24 | 74.10% ± 0.44% | 0.99% ± 0.45% |
| | | CND | 13.00 ± 5.39 | -6.60 ± 5.81 | 73.25% ± 1.80% | 1.85% ± 2.19% |
| | 9.995% | KPA | 11.60 ± 5.13 | -24.40 ± 11.01 | 76.88% ± 0.91% | 2.29% ± 0.83% |
| | | PC | 21.00 ± 9.14 | -15.00 ± 9.46 | 77.85% ± 0.85% | 1.31% ± 0.71% |
| | | CND | 16.00 ± 4.64 | -20.00 ± 8.97 | 78.04% ± 0.86% | 1.13% ± 0.82% |

(c) NEWS

Table 2: Mini-WebVision and Tiny-ImageNet (symmetric label noise) early stopping comparison.

| dataset | metric | $e$ | $\Delta e$ | $a$ | $\Delta a$ |
|---|---|---|---|---|---|
| | KPA | 8 | -296 | 26.96 | 47.96 |
| Mini-WebVision | PC | 375 | 71 | 71.68 | 3.24 |
| | CND | 335 | 31 | 73.32 | 1.60 |
| | KPA | 5.00 | -79.00 | 38.54 | 8.51 |
| Tiny-ImageNet (10%) | PC | 78.00 | -6.00 | 43.50 | 3.55 |
| | CND | 4.00 | -80.00 | 34.97 | 12.08 |
| | KPA | 5.00 | -8.00 | 34.93 | 8.67 |
| Tiny-ImageNet (20%) | PC | 92.00 | 79.00 | 40.37 | 3.23 |
| | CND | 6.00 | -7.00 | 37.19 | 6.41 |
| | KPA | 5.00 | -10.00 | 34.45 | 6.46 |
| Tiny-ImageNet (30%) | PC | 99.00 | 84.00 | 35.64 | 5.27 |
| | CND | 7.00 | -8.00 | 36.92 | 3.99 |
| | KPA | 5.00 | -15.00 | 24.01 | 12.94 |
| Tiny-ImageNet (40%) | PC | 99.00 | 79.00 | 28.81 | 8.15 |
| | CND | 13.00 | -7.00 | 34.27 | 2.69 |

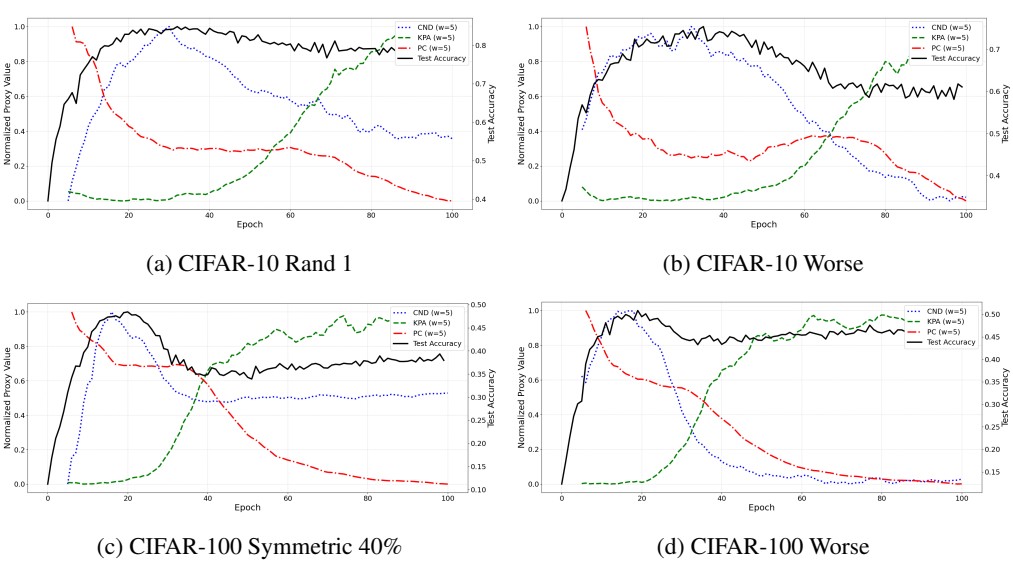

(a) CIFAR-10 Rand 1

(b) CIFAR-10 Worse

(c) CIFAR-100 Symmetric 40%

(d) CIFAR-100 Worse

Figure 4: Representative training curves showing the dynamics of CND, PC, and KPA under different noise settings.

## LLM USAGE

LLMs were used only to improve clarity and readability of the text, as well as to speed up the coding process. They were not used for research ideation, experimental design, analysis, or generation of technical content. All scientific contributions are solely those of the authors. After using the tool, the authors carefully reviewed and edited the content, and take full responsibility for the final manuscript.

**Reproducibility Statement.** We provide an *anonymous* supplementary `.zip` archive containing the full source code, ready-to-run scripts, and experiment configuration files (the JSONs under `programs/simulations/`) used to generate all results in the paper. The archive includes a `README.md` with environment setup, commands to launch experiments, and instructions to download/prepare data. Dataset sources and preprocessing details are documented in Appendix C; training schedules, hyperparameters, and metric settings are specified in Section 5 and Appendix D; the early-stopping procedure is given in Algorithm 1 (Appendix B); and additional qualitative/quantitative results and plots are provided in Appendix E. A de-anonymized public repository will be released upon acceptance.

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

# A  STATISTICAL ASSESSMENT OF CLASS-CONDITIONAL NEURAL DISTRIBUTIONS

To quantify the proportion of neurons exhibiting multimodal pre-activation distributions, we applied the Kruskal–Wallis non-parametric test (Hollander et al., 2013). This test generalizes the Mann–Whitney U test to more than two groups and ranks data without assuming normality, making it well suited for analyzing neural activations. Table 3 reports the percentage of neurons with statistically significant differences ($p < 0.05$) across class-conditional distributions. These results indicate that most neurons follow a multimodal class-conditional mixture model. For completeness, apart to CIFAR-10 with Resnet-9, we also include results obtained with a 4 layers fully connected networks trained on MNIST and Fashion-MNIST, both with dropout and without.

Table 3: Significant Neurons per Layer (Kruskal non-parametric Statistical Test)

| Dataset | Dropout | Label Noise | Sig_Neurons by Layer Index | | | |
|---|---|---|---|---|---|---|
| | | | **(0)** | **(1)** | **(2)** | **(3)** |
| CIFAR10 | 0.0 | clean_label | 1.00 | 1.0 | 1.0 | 1.0 |
| | 0.0 | random_label1 | 1.00 | 1.0 | 1.0 | 1.0 |
| | 0.0 | worse_label | 0.98 | 1.0 | 1.0 | 1.0 |
| | 0.5 | clean_label | 1.00 | 1.0 | 1.0 | 1.0 |
| | 0.5 | random_label1 | 1.00 | 1.0 | 1.0 | 1.0 |
| | 0.5 | worse_label | 0.98 | 1.0 | 1.0 | 1.0 |
| FashionMNIST | 0.0 | 0.0 | 1.00 | 1.0 | 1.0 | 1.0 |
| | 0.0 | 0.2 | 1.00 | 1.0 | 1.0 | 1.0 |
| | 0.0 | 0.4 | 1.00 | 1.0 | 1.0 | 1.0 |
| | 0.0 | 0.6 | 1.00 | 1.0 | 1.0 | 1.0 |
| | 0.5 | 0.0 | 1.00 | 1.0 | 1.0 | 1.0 |
| | 0.5 | 0.2 | 1.00 | 1.0 | 1.0 | 1.0 |
| | 0.5 | 0.4 | 1.00 | 1.0 | 1.0 | 1.0 |
| | 0.5 | 0.6 | 1.00 | 1.0 | 1.0 | 1.0 |
| MNIST | 0.0 | 0.0 | 1.00 | 1.0 | 1.0 | 1.0 |
| | 0.0 | 0.2 | 1.00 | 1.0 | 1.0 | 1.0 |
| | 0.0 | 0.4 | 1.00 | 1.0 | 1.0 | 1.0 |
| | 0.0 | 0.6 | 1.00 | 1.0 | 1.0 | 1.0 |
| | 0.5 | 0.0 | 1.00 | 1.0 | 1.0 | 1.0 |
| | 0.5 | 0.2 | 1.00 | 1.0 | 1.0 | 1.0 |
| | 0.5 | 0.4 | 1.00 | 1.0 | 1.0 | 1.0 |
| | 0.5 | 0.6 | 1.00 | 1.0 | 1.0 | 1.0 |

# B  EARLY STOPPING ALGORITHM AND SETTINGS

To implement the proposed early stopping strategies, we developed a unified algorithm that can be applied to different metrics (PC, KPA, and CND). The method tracks the smoothed trajectory of each metric using a moving average and identifies local minima or maxima depending on the metric type. Training proceeds until no new best value is observed within a predefined patience window, after which the model parameters from the best epoch are returned. The full procedure is presented in Algorithm 1.

# C  DATASETS

We conducted experiments on several benchmark datasets that differ in modality, scale, and complexity.

**CIFAR-10** consists of 60,000 natural images of size $32 \times 32$ pixels equally distributed across 10 classes, with 50,000 training samples and 10,000 test samples.

---

**Algorithm 1:** Early Stopping Based on Local Minima/Maxima

---

**Input:** patience $p$, smoothing window $k$, metric type $M \in \{\text{PC}, \text{KPA}, \text{CND}\}$
**Output:** best parameters $\theta^*$ and best epoch $t^*$
Initialize $\theta \leftarrow \theta_0, t \leftarrow 0, wait \leftarrow 0, \theta^* \leftarrow \theta, t^* \leftarrow 0$

$$v_{\text{best}} \leftarrow \begin{cases} +\infty, & M \in \{\text{PC}, \text{KPA}\} \\ -\infty, & M = \text{CND} \end{cases}$$

**while** *wait* $< p$ **do**

    Update $\theta$ for one epoch; $t \leftarrow t + 1$
    $X_t \leftarrow$ raw metric value at epoch $t$
    **if** $t < k$ **then**
        **continue**
                          `// fill the window before evaluating`

    $S_t \leftarrow \frac{1}{k} \sum_{j=t-k+1}^{t} X_j$
                         `// moving average over the last k epochs`
    **if** $(M \in \{\text{PC}, \text{KPA}\} \wedge S_t < v_{best}) \vee (M = \text{CND} \wedge S_t > v_{best})$ **then**
        $v_{\text{best}} \leftarrow S_t; wait \leftarrow 0; \theta^* \leftarrow \theta; t^* \leftarrow t$
    **else**
        $wait \leftarrow wait + 1$

**return** $\theta^*, t^*$

---

**CIFAR-100** shares the same format but contains 100 classes with 600 images each (500 for training and 100 for testing). Its larger number of classes and finer granularity make it a more challenging benchmark for studying generalization and robustness under noisy labels.

**CIFAR-10N** and **CIFAR-100N** are real-world noisy variants of CIFAR-10 and CIFAR-100 introduced by Wei et al. (2022). Instead of synthetically flipping labels at random, these datasets provide human-annotated labels collected via Amazon Mechanical Turk. Each image has multiple annotations, and majority voting yields a "clean" label, while the raw crowd-sourced labels introduce realistic label noise reflecting human errors and ambiguities. CIFAR-10N contains five different noisy label sets (including *aggregate*, *random 1*, *random 2*, *worse*, and *aggre* variants), whereas CIFAR-100N provides a single noisy label set. These benchmarks are widely used to evaluate robustness under *instance-dependent*, human-like noise.

**Mini-WebVision** (Li et al., 2017) is a web-scale benchmark derived from the first 50 ILSVRC categories of the WebVision dataset. It contains approximately 66k training images collected from Flickr and Google Image Search, accompanied by noisy meta-data such as titles and user-generated tags, and 13k test images sourced from the corresponding ImageNet validation set. Compared to CIFAR-style datasets, Mini-WebVision features higher-resolution images, substantial class imbalance, and naturally occurring label noise introduced by imperfect web search queries.

**Tiny-ImageNet** mirrors the ImageNet taxonomy at reduced scale and resolution: it comprises 200 classes with 500 training images, 50 validation images, and 50 test images per class, all resized to $64 \times 64$ pixels. Despite its smaller footprint, the dataset retains ImageNet's fine-grained semantics.

Finally, the **20 Newsgroups (NEWS20)** dataset (Yu et al., 2019; Kiryo et al., 2017) is a widely used text classification benchmark consisting of approximately 20,000 documents partitioned into 20 balanced categories, such as politics, science, sports, and technology. Each document is represented as raw text, which we encode using pretrained GloVe embeddings. Unlike the image datasets, NEWS20 introduces a natural language modality with high-dimensional and sparse features, making it a complementary testbed for analyzing the behavior of early stopping metrics under label noise.

## D   Simulation Parameters

Table 4 summarizes the training settings used in all experiments. For CND, we focus on the last hidden layer, as it consistently exhibits the strongest correlation with generalization. We estimate CND using a histogram-based plug-in estimator of differential Shannon entropy. Specifically, for

each neuron, we collect its pre-activations over the training set, discretize the activation range into $B$ equal-width bins, and compute both class-conditional and marginal histograms. Entropies are then approximated using the standard histogram correction (Hall & Morton, 1993; Beirlant et al., 1997),

$$\hat{h}(X) \approx - \sum_{i=1}^{B} p_i \, \log\left(\frac{p_i}{\Delta_i}\right), \tag{7}$$

where $p_i$ is the empirical probability mass in bin $i$ and $\Delta_i$ is the bin width. The JSD for each neuron is computed as the entropy of the marginal histogram minus the average entropy of the class-conditional histograms. We selected this estimator primarily for its computational efficiency, which is crucial given the repeated entropy evaluations required during training. More accurate alternatives (e.g., kNN- or KDE-based entropy estimators) were excluded due to their significantly higher per-epoch cost, which made them impractical for online tracking across all neurons and layers. In contrast, the histogram plug-in estimator provides a fast, vectorizable approximation that still captures the relevant dynamics.

When aggregating neuron-wise CND values, we report the 90th percentile across neurons in the last hidden layer. This statistic proved more stable and robust to outliers than the mean, and it better aligned with the point of maximum generalization. High-CND neurons tend to behave more erratically once memorization begins, and Appendix G shows through pruning experiments that these neurons are also the ones most critical for generalization.

Finally, in Table 4, *Seeds* denotes the number of independent runs (with different random initializations). Results are reported as mean $\pm$ standard deviation when Seeds $> 1$, and as a single run otherwise.

Table 4: Simulation parameters for all experimental setups.

| Parameter | NEWS20 | CIFAR-10/100 | Mini-WebVision | Tiny-ImageNet |
|---|---|---|---|---|
| Network | 3-layer MLP | Preact-ResNet18 (He et al., 2016) | ResNet-50 | ResNet-50 |
| Epochs | 200 | 100 | 400 | 100 |
| Optimizer | SGD (0.9 mom., Nesterov) | SGD (0.9 mom., Nesterov) | SGD (0.9 mom., Nesterov) | SGD (0.9 mom., Nesterov) |
| Init. LR | 0.001 | 0.1 | 0.5 | 0.1 |
| LR schedule | – | – | Multi-step (200,300) | Multi-step (50,80) |
| Weight decay | $5 \times 10^{-4}$ | $1 \times 10^{-5}$ | $1 \times 10^{-5}$ | $1 \times 10^{-5}$ |
| Loss | Cross-entropy | Cross-entropy | Cross-entropy | Cross-entropy |
| Batch size | 128 | 256 | 32 | 32 |
| Seeds | 10 | 5 | 1 | 1 |

# E  EARLY STOPPING METRICS PLOT COMPARISON

In Figures 5 and 6, we present training curves from selected simulations that illustrate how the proposed early stopping metrics behave under different label noise conditions and varying moving average window sizes. This analysis highlights the sensitivity of these proxies to this crucial hyperparameter.

Starting with CIFAR-10 in Figure 5, we observe that the PC metric does not consistently exhibit the "label wave" pattern described by Yuan et al. (2023a). This behavior becomes clearly visible only in high-noise scenarios—such as 5b, 5c, 5d, and 5h—where a well-defined local minimum emerges. In contrast, when the noise is moderate—e.g., in 5f and 5g—the local minimum is less distinct, and inadequate tuning of the moving average window may result in missing the generalization peak. In these cases, the noise still substantially degrades performance. Under low-noise conditions—such as 5a and 5e—the PC metric fails to identify any meaningful transition, which aligns with the absence of a noticeable drop in test accuracy. Regarding the behavior of CND, the curves generally align well with test accuracy, except in 5d, where we hypothesize that the extremely high label noise prevents the emergence of clearly class-dependent distributions. Notably, in 5e, where no significant decline in test accuracy occurs, the CND curve remains stable, accurately tracking generalization performance without displaying a spurious local maximum. A key limitation of CND is its inability to reflect the plateau in test accuracy that sometimes follows the memorization phase. During this plateau, accuracy may fluctuate or temporarily improve, yet the CND signal remains insensitive to these changes. By contrast, the KPA metric is one of the more robust estimators, remaining effective even

in high label noise scenarios such as 5d. However, it tends to stop training too early due to its reliance on the first-local-minimum principle.

Turning to the CIFAR-100 results in Figures 6, the PC metric fails to identify a local minimum even in highly noisy settings, as seen in 6d. In the other two cases, 6b and 6c, improper selection of the window size may also lead to missing the local maxima. In 6a, the CND metric fails because it detects an early local maximum, even though the highest test accuracy is achieved later in the training.

Analyzing the NEWS plots, we first note that the training process starts with a pretrained GloVe embedding; hence, the initial CND can be higher than the subsequent local maxima. Nevertheless, despite this difference, the CND still exhibits a local maximum at the point where generalization peaks. In contrast, all the other metrics display high instability, and PC fails to detect the onset of memorization under low noise conditions (Figure 7a), even though this noise level is not negligible and memorization does affect the network's generalization ability.

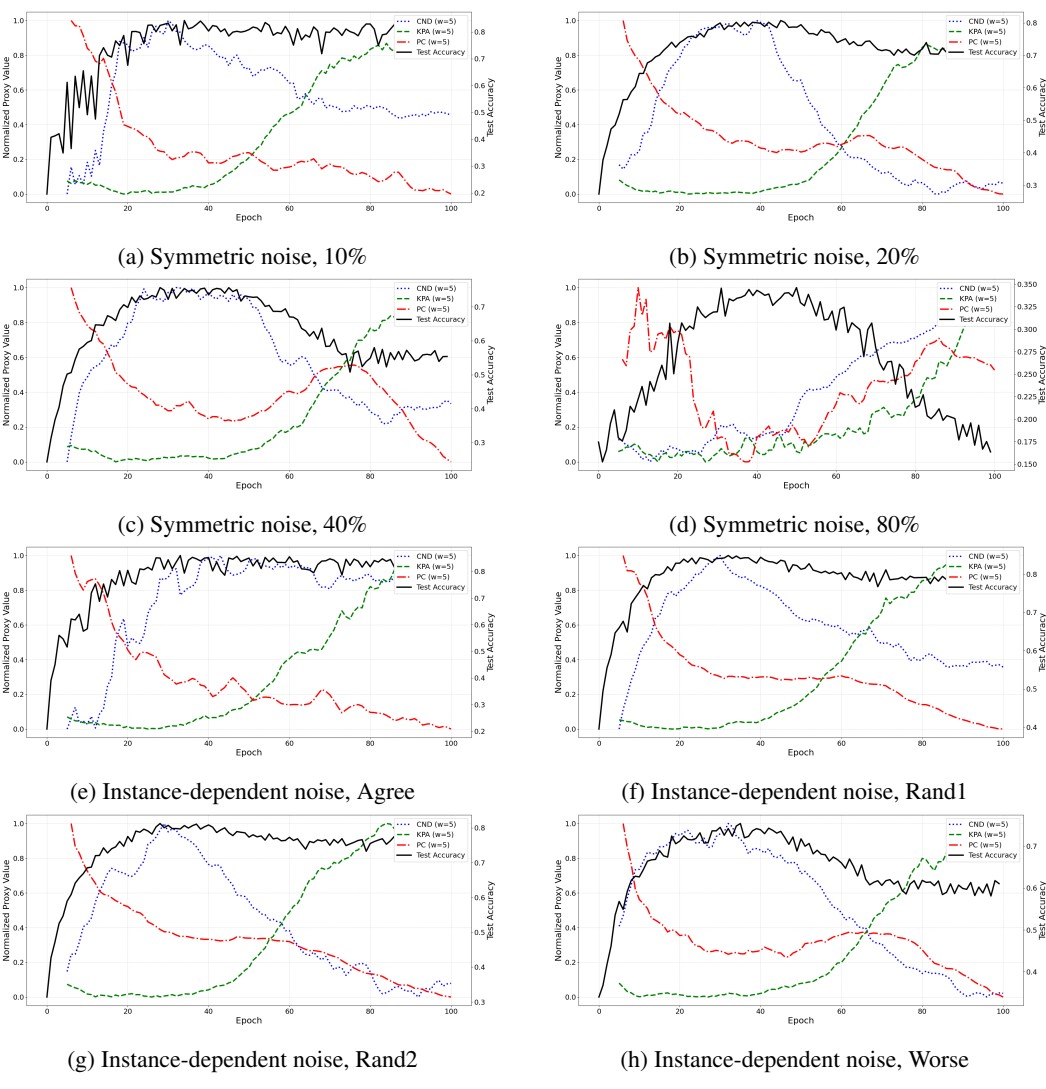

(a) Symmetric noise, 10%    (b) Symmetric noise, 20%

(c) Symmetric noise, 40%    (d) Symmetric noise, 80%

(e) Instance-dependent noise, Agree    (f) Instance-dependent noise, Rand1

(g) Instance-dependent noise, Rand2    (h) Instance-dependent noise, Worse

Figure 5: Training curves with early stopping proxies on CIFAR-10 under different types and intensities of label noise.

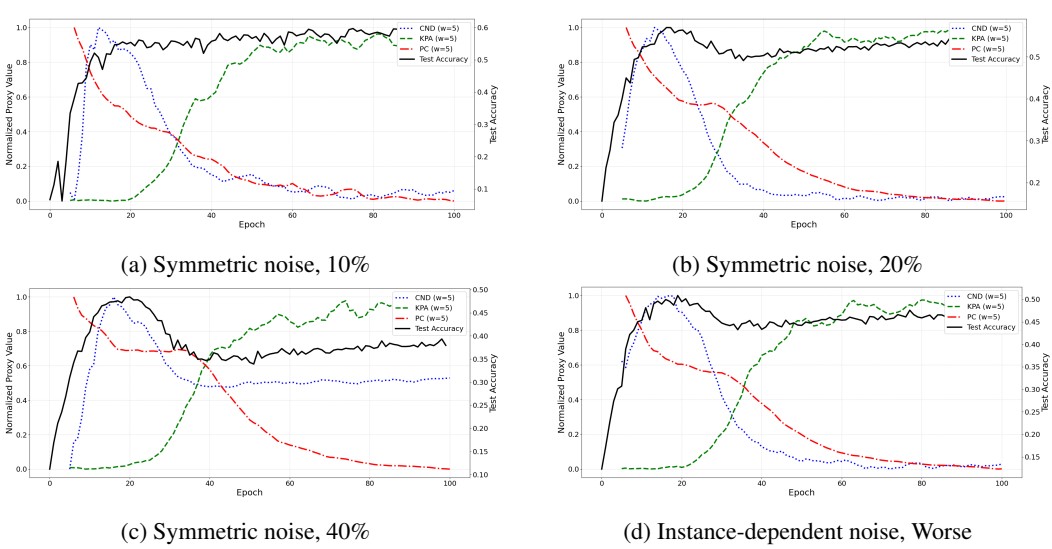

Figure 6: Training curves with early stopping proxies on CIFAR-100 under different types and intensities of label noise.

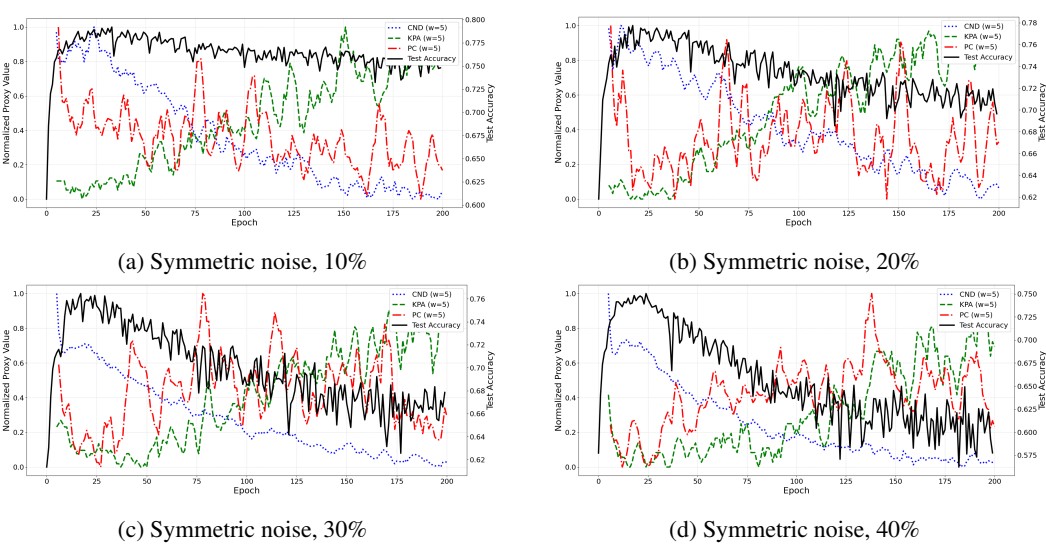

Figure 7: Training curves with early-stopping proxies on NEWS dropout experiment at different simulation levels.

## F  CND over a Clean Training Subset under High Label Noise

In the main work, CND is computed over the entire training set, including samples affected by label noise. A fundamental advantage of this approach is that it eliminates the need for a clean validation set, allowing the method to be applied in scenarios where ground truth data is unavailable. However, as noted in the main text, CND becomes unstable when the level of label noise is particularly high. To address this extreme regime, we evaluate a solution based on computing CND using only a small, known clean subset of the training data. Crucially, unlike a traditional validation set, this clean subset is not held out; it remains part of the training loop to prevent the loss of valuable information. The subset acts solely as an anchor for the divergence estimation. We evaluated this approach on CIFAR-10 with 60% symmetric noise, comparing the standard CND (estimated from the fully noisy set) against a variant computed over a 10% clean subset. As summarized in Table 5, the standard CND yields unstable results with high variability in this regime. In contrast, the clean-subset anchor drastically stabilizes the stopping point. This method halts just 3.33 epochs on average away from the optimum ($\Delta e$) and limits the accuracy drop ($\Delta a$) to 1.36%, compared to an 11.82% loss when monitoring the noisy set. This demonstrates that CND remains effective even at extreme noise rates, provided a modest subset of trusted labels is available to drive the divergence estimate.

Table 5: Comparison between the original CND (monitoring metrics on the noisy training set) and the variant where CND is computed on a 10% clean subset injected into training. CIFAR-10 symmetric 60% results

| noise type | label noise ratio | metric | $e$ mean ± std | $\Delta e$ mean ± std | $a$ mean ± std | $\Delta a$ mean ± std |
|---|---|---|---|---|---|---|
| | | KPA | 20.67 ± 3.21 | -3.67 ± 6.03 | 69.75% ± 0.45% | 2.41% ± 0.51% |
| | | PC | 8.00 ± 2.65 | -16.33 ± 6.51 | 59.78% ± 6.22% | 12.37% ± 5.39% |
| | | CND (train set only) | 37.80 ± 21.78 | -8.40 ± 23.22 | 53.17% ± 12.28% | 11.82% ± 13.09% |
| symmetric | 60% | CND (10% clean subset) | 21.00 ± 2.00 | -3.33 ± 4.93 | 70.79% ± 1.15% | 1.36% ± 1.36% |

## G  Neuron Pruning

Here we question whether it is possible to determine which information is contained in which neurons based on their CND values. To do this, we applied a threshold pruning strategy based on CND. First, we compute the layer-specific CND:

$$T_l^{(q)} = Q_q\big(\{\text{CND}_l[i] \mid i = 1, \ldots, N_l\}\big),  \tag{8}$$

Let $l$ denote the layer index, $Q_q$ the $q$-th quantile function, and $N_l$ the number of neurons in layer $l$. The value $T_l^{(q)}$ represents the pruning threshold defined by a specific quantile of the $\text{CND}_l$ distribution. Pruning consists of removing all neurons in the set $\mathcal{P}_l$. Thresholds are computed layer-wise, considering the consistent behavior of neurons within each layer.

We then investigated how prediction performance changes under different pruning levels, beginning with the removal of neurons with low CND values and repeating the experiment by pruning those with high CND values. These experiments were conducted on networks exhibiting memorization. In both cases, results were compared with random pruning—where the same number of neurons is removed randomly—to assess the effectiveness of the CND-based selection criterion.

In Figures 8, 9, and 10, we plot test accuracy alongside training accuracy—separated into clean and corrupted samples—to analyze how networks behave under different levels and types of pruning, for both label-noise-memorized and clean instances. We apply two pruning strategies: one based on CND and another based on randomness, to assess whether CND-based pruning is truly effective or if its effect is equivalent to simply deactivating a fixed percentage of neurons.

With low-level pruning, we observe that accuracy on corrupted samples begins to drop as soon as neurons are pruned. However, this drop is roughly equivalent to that seen with random pruning, suggesting that CND-based pruning does not precisely localize neurons responsible for memorizing noisy labels. In contrast, a clear difference emerges in clean training and test accuracy. Specifically,

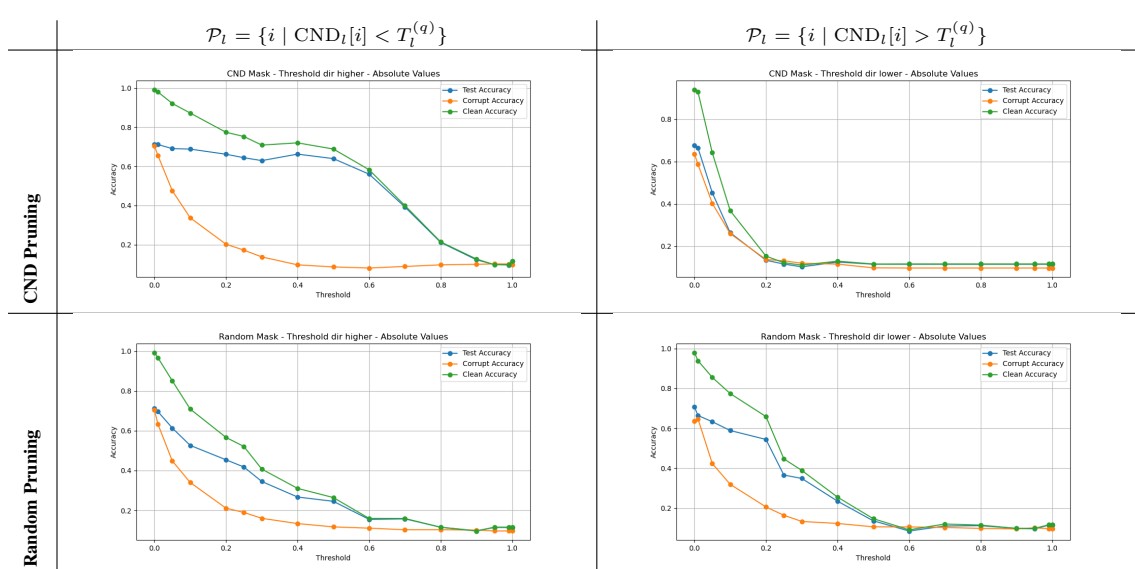

Figure 8: MNIST (FC DNN, 0.4 symmetric label noise). Rows: CND-based pruning (top) vs. random (bottom). Columns: pruning low-CND neurons (left) vs. high-CND neurons (right). High-CND pruning sharply hurts test accuracy; low-CND pruning mostly drops clean-train accuracy while test accuracy holds longer.

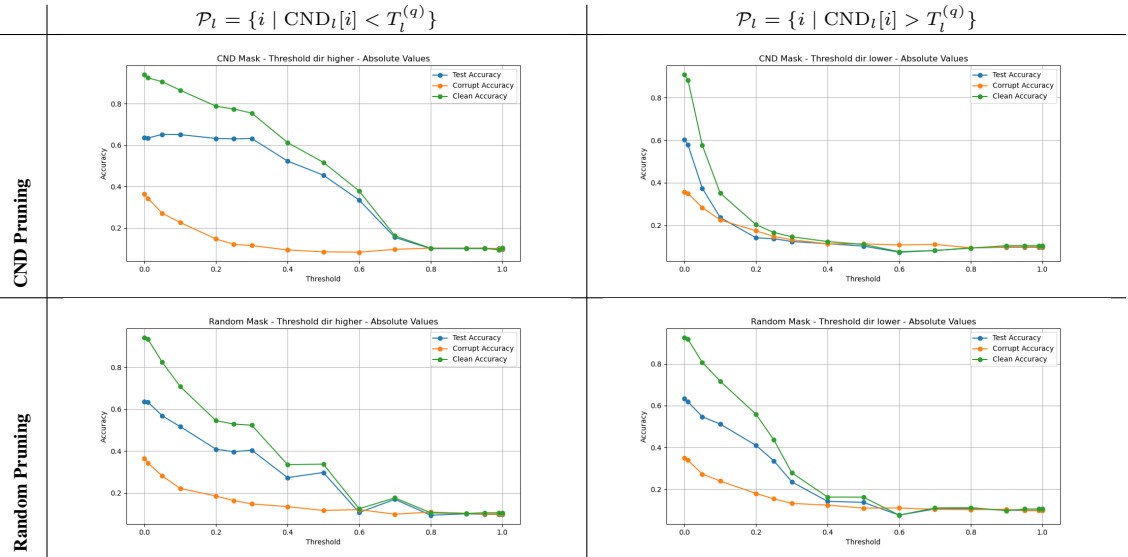

Figure 9: Fashion-MNIST (FC DNN, 0.6 symmetric label noise). Same layout as Fig. 8.

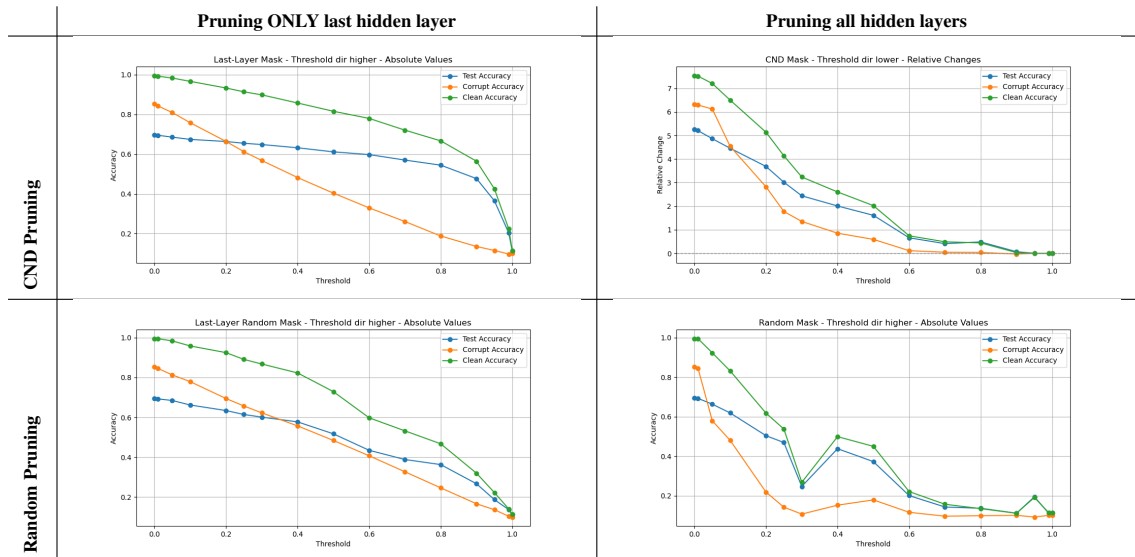

Figure 10: SVHN (LeNet, 0.2 symmetric label noise). Columns: pruning only the last hidden layer (left) vs. all hidden layers (right); rows: CND-based pruning (top) vs. random (bottom). High-CND pruning harms test accuracy more than random, even when limited to the last layer; pruning all layers shows extra loss from conv units but the same CND-vs.-random gap. Low-CND pruning mainly affects clean-train accuracy.

when pruning low-CND neurons, test accuracy remains stable at first, while clean training accuracy declines immediately. This behavior is not observed with random pruning, where both metrics degrade simultaneously. These results suggest that neurons with high CND values contribute more to generalization, whereas memorization is more diffusely distributed across the network and not well captured by CND.

Furthermore, when pruning high-CND neurons, generalization performance deteriorates faster than in the random pruning case. This confirms that high-CND neurons are critical for generalization. In the final case (Figure 10)—the only scenario involving a convolutional architecture—pruning convolutional layers significantly impacts performance. To isolate this effect, we restrict pruning to the final fully connected hidden layer. Here again, we observe that high-CND neurons are more important for generalization, as evidenced by the more gradual performance degradation compared to random pruning.

