# OpenReview forum: "Class-Conditional Neuron Pre-Activation Divergence to rule out validation set in label noise early stopping"
_ICLR.cc/2026/Conference — Submitted to ICLR 2026_

### Official Review · Reviewer_dezB · 2025-10-15

**Soundness:** 2
**Presentation:** 2
**Contribution:** 3
**Rating:** 4
**Confidence:** 5

**Summary:**

This work studies the relation between neuron pre-activation distributions and label-noise memorization. The authors hypothesize that the class-dependent structure in these distributions weakens as memorization sets in.

To capture this transition, they introduce a metric, Class-Conditional Neuron Pre-Activation Divergence (CND). Based on the observation that CNDs of the last hidden layer peak in alignment with generalization, they propose a validation-free early stopping criterion.

Their experiments reveal that CND outperforms baseline method.

**Strengths:**

1.  The work proposes a novel approach by linking neuron pre-activation distributions to label noise memorization, offering an interesting perspective on identifying the generalization peak.
2.  The core hypothesis is intuitive and well-grounded.
3.  The proposed CND metric demonstrates strong performance against PC in their settings.

**Weaknesses:**

My overall view is that this appears to be a first paper written by a very junior author and has a considerable number of flaws. After discussing the paper, I will provide some suggestions for junior authors on academic writing, offered with my best wishes. (I have read the paper in its entirety. These suggestions are not necessarily for you to revise in the rebuttal, but are fundamental advice for your future work).

Returning to the paper's content, I think the topic and the idea are interesting and have some insight. At a time when many choose quick a+b+c papers on LLMs or somehow ChatGPT can do something we are already known as an ACL-style entry point, we should encourage ML researchers like this, who are genuinely interested in discussing a small and refined topic about deep networks themselves.
Out of encouragement, I am giving a borderline reject before the rebuttal and will keep an open mind.

I encourage the authors to get more research training (in reading literature, experimental design, and paper writing) before submitting to a top-tier conference again.

**Questions:**

**Major:**

1. I could not find the details of the experimental setup in the main text. The authors state, "For details about the datasets, see Appendix C, and for the simulation settings, see Appendix D," yet I was unable to find the appendix (it was not in the Supplementary Material either). Even if it were provided, this critical information should be included in the main body of the paper for self-containment.

2. The choice of noise settings is not well-justified. For instance, why is the maximum noise rate for CIFAR-100 only 40%? Why was only one type of synthetic noise evaluated? Furthermore, the baseline comparison is incomplete; it lacks a crucial comparison against standard early stopping with a small validation set.

3. What is PMF in Table 1(c)? The term is used without any prior definition in the text.

4. The claimed equivalence from Eq. (4) to Eq. (5) does not seem to hold under the given context. It appears to depend on the relationship between the true marginal distribution P(z) and the uniform mixture M(z) used in the JSD definition (I do not really understand it, although I have try my best to follow their flow).

5. While the paper links CND to the phenomenon of memorization, it lacks a deeper analysis of why memorizing noise would cause the class-conditional distributions to collapse toward the marginal distribution. The authors should provide at least a self-consistent reason for this core claim.

**Minor:**

1. On line 249, the index [i] is used. However, only the index [j] has been previously defined. It is unclear where i originates from.

2. The method for estimating the continuous probability density functions from data needs to be described more clearly.

**Suggestions:**

(Not necessarily for you to revise in the rebuttal)

1. The paper has many short paragraphs, some consisting of only a single line ("orphans"). These should be consolidated into more developed paragraphs to improve readability and flow. A paragraph should ideally contain more than just a few sentences.

2. The equation around line 333 is unnumbered. The caption for Figure 1 contains the phrase, "...highlighting the multimodal behavior induced by backdoor triggers." This is confusing.

3. Placing all the experimental tables on Page 8, a page with no accompanying text, is not appropriate. Tables and figures should be placed near the text that discusses them.

4. The paper does not make full use of the page limit, suggesting a lack of depth. A more thorough evaluation is expected for a top-tier venue. For example, what ablation studies are missing? The authors should consider ablating the CND aggregation rule (e.g., mean vs. quartile) or the specific layer chosen.

5. The figures themselves are very large (maybe too large to leave space in paper for more information), yet the font size for labels and text within them is too small, which harms readability.

6. Have you considered adding a simple, illustrative diagram on page 2 to visually summarize your key finding? This could be referenced in the introduction and would significantly help readers quickly grasp the core concept of your work.

---

As a final suggestion, I recommend you carefully read the following papers to study their writing, visualization, experimental settings, and overall paper layout:

[1] Toneva, Mariya, et al. "An Empirical Study of Example Forgetting during Deep Neural Network Learning." International Conference on Learning Representations, 2019.

[2] Pleiss, Geoff, et al. "Identifying mislabeled data using the area under the margin ranking." Advances in Neural Information Processing Systems 33 (2020): 17044-17056.

[3] Li, Junnan, Richard Socher, and Steven CH Hoi. "DivideMix: Learning with Noisy Labels as Semi-supervised Learning." International Conference on Learning Representations, 2020.

[4] Frankle J, Carbin M. The Lottery Ticket Hypothesis: Finding Sparse, Trainable Neural Networks[C]//International Conference on Learning Representations, 2019.

---

> ### Author Response · Authors · 2025-11-20
>
> We thank the reviewer for the thorough feedback and constructive suggestions.
>
> ### Major 1
>
> The missing appendix will be included. Key experimental details will also be moved into the main text to improve self-containment.
>
> ### Major 2
>
> We acknowledge inconsistencies in noise settings across datasets. We will expand CIFAR-100 experiments to include the missing settings. The standard early-stopping baseline was not included because, in *S. Yuan, L. Feng, and T. Liu, “Early Stopping Against Label Noise Without Validation Data,” 2024*, the PC method was already compared against it, demonstrating the advantages of a validation-free approach. Nevertheless, we will also include this baseline in the next revision.
>
> ### Major 3
>
> The reference to "PMF" was an artifact from automated table generation and has been removed.
>
> ### Major 4
>
> We thank the reviewer for pointing this out. The transition from Eq. (4) to Eq. (5) was not sufficiently justified in the submitted version. In the revision, we will make this explicit and include a short derivation. In particular, the mixture used in the JSD is
>
> $M_j^{(l)} = \sum_{y \in \mathcal{Y}} \pi_y \, P(z_j^{(l)} \mid y)$
>
> This equals the marginal distribution $P(z_j^{(l)})$ **only if** the class prior matches the JSD weights, i.e., $P(y) = \pi_y$ for all $y \in \mathcal{Y}$. We will state this condition explicitly and expand the derivation to avoid ambiguity.
>
> ### Major 5
>
> This is a crucial point we are trying to address. RIght now, we will try to improve it adding ablations to clarify the mechanism behind collapse:
> - **Information bottleneck and memorization:** an ablation connecting collapse dynamics to information bottleneck behavior and noisy-label memorization.
> - **Label vs. feature noise:** an ablation showing that the characteristic CND collapse arises with **label-noise memorization** but not under feature/input noise.
>
> ### Minor 1
>
> The indexing typo has been corrected.
>
> ### Minor 2
>
> Density estimation uses a histogram-based approximation of differential entropy for computational efficiency. A detailed description will be available in the appendix.
>
> ### Additional Suggestions
>
> We appreciate the reviewer’s guidance and will address as many of these points as possible in the revised version.

---

> > ### Comment · Reviewer_dezB · 2025-11-26
> >
> > Thank you for your response. I acknowledge and appreciate the efforts you have put into the rebuttal. However, I believe the manuscript, in its current state, has not yet reached the acceptance standard for ICLR.
> >
> > Out of encouragement for the authors, I will maintain my current score. I strongly suggest further refining the paper. After these revisions, I encourage you to submit this work to venues with similar topics, such as UAI or AISTATS.

---

### Official Review · Reviewer_3CiR · 2025-10-29

**Soundness:** 3
**Presentation:** 2
**Contribution:** 2
**Rating:** 2
**Confidence:** 4

**Summary:**

This paper addresses the problem of early stopping under label noise without using a clean validation set. The authors propose a novel metric, Class-Conditional Neuron Pre-Activation Divergence (CND), which measures the divergence between class-conditional and marginal distributions of neuron pre-activations. They also show that when a network begins to memorize noisy labels, the class-specific neuron pre-activation collapses toward the marginal distribution.

Based on this observation, the authors design a validation-free early stopping criterion to stop training when the last-layer CND reaches its peak. Experiments on multiple benchmarks (MNIST, CIFAR-10/100, CIFAR-10N, and NEWS) demonstrate that CND consistently identifies optimal stopping epochs more accurately than prior validation-free methods such as Prediction Changes (PC) and Known Polluted Accuracy (KPA), thereby improving the performance.

**Strengths:**

1. The paper introduces a novel perspective for early stopping through neuron pre-activation distributions.
2. The authors propose Class-Conditional Neuron Pre-Activation Divergence (CND) to measure the divergence between class-conditional and marginal pre-activation distributions.
3. The authors conduct experiments on several datasets to verify the effectiveness of the proposed method. Compared to the previous method, the proposed method can better identify the early stopping epoch.

**Weaknesses:**

1. The experiments are mainly conducted on small or medium-scale datasets such as MNIST and CIFAR. Evaluating the proposed method on larger and more realistic datasets (e.g., Clothing1M, WebVision) would better demonstrate its scalability and practical utility under real-world settings.
2. The failure cases at very high noise rates (≥60%) are mentioned but not deeply analyzed. Additional discussion on why CND fails would improve completeness.
3. The comparison mainly focuses on validation-free baselines. It would be better to compare with other baselines published in recent years.
4. The current version appears incomplete. Multiple cross-references to appendices are unresolved, and the corresponding supplementary materials also do not include the Appendix.

**Questions:**

1. The meanings of the x-axis and y-axis in Figure 1 are not clearly described. Could the authors provide more details about them?
2. Why are the lines on Subfigure (f) more than those on the other Subfigures?

---

> ### Author Response · Authors · 2025-11-20
>
> Thank you for the detailed analysis and valuable suggestions.
>
> ### Clarification
>
> Known Polluted Accuracy (KPA) is a new metric introduced in this work as an additional validation-free baseline, not a previously published method. Prediction Changes (PC) is the previously published validation-free baseline in the literature. This distinction will be clarified in the paper to avoid confusion.
>
> ### Weakness 1
>
> We agree that demonstrating scalability is essential. Experiments on Clothing1M and WebVision are in progress and will be included in the revised manuscript. Early qualitative results indicate comparable collapse dynamics in these larger settings.
>
> ### Weakness 2
>
> This behavior arises because CND is computed over the training dataset, meaning we observe $p(z \mid \tilde{y})$ rather than $p(z \mid y)$, where $\tilde{y}$ denotes the observed label, which may include misannotations. At high symmetric noise levels ($\mu \ge 60\%$), the conditional density becomes dominated by incorrect labels. In practice, the observed distribution may be described as a mixture:
>
> $p(z \mid \tilde{y}) = (1 - \mu)p(z \mid y) + \mu p(z)$
>
> As a result, the onset of memorization becomes difficult to distinguish from the normal behavior of the CND under heavy corruption. We will expand the discussion of this limitation and clarify this point in the revised paper.
>
>
> ### Weakness 3
>
> Since this work focuses specifically on validation-free early stopping rather than designing a full noisy-label training pipeline, broader baselines were not compared in terms of end-to-end performance. We will clarify this scope and emphasize that CND is complementary and can be combined with such methods.
>
> ### Weakness 4
>
> The missing appendix will be reinstated, and figure captions will be updated for clarity.
>
> ### Question 1
>
> The x-axis represents neuron pre-activation values, and the y-axis represents estimated density values. This clarification is now included.
>
> ### Question 2
>
> Subfigure (f) includes 100 curves because CIFAR-100 contains 100 classes. This is now stated explicitly in the caption.

---

### Official Review · Reviewer_skds · 2025-10-29

**Soundness:** 3
**Presentation:** 2
**Contribution:** 3
**Rating:** 6
**Confidence:** 3

**Summary:**

The paper proposes a novel metric called Class-Conditional Neuron Pre-Activation Divergence (CND), which measures the divergence between class-conditional and marginal pre-activation distributions of neurons. This metric is used to develop a validation-free early stopping criterion to mitigate label noise in deep learning models. The method claims to be particularly effective in datasets with multiple classes and lower noise levels, offering better generalization compared to existing methods.

**Strengths:**

1. The CND metric offers a new perspective on how neuron activations evolve during training, providing a novel tool for detecting memorization due to noisy labels.
2. The paper is well-structured, with clear explanations of the key concepts and results.

**Weaknesses:**

1. Some details are not clearly explained. The implementation details of the figures and formulas in the paper, such as the distinction between memorization and generalization in Figure 2, need more detailed explanations. This is particularly important for readers who may not have related background knowledge.
2. Limited experimental scope. The paper only evaluates on very small models using a relatively single-type dataset. Larger models generally have better noise tolerance, and the paper does not demonstrate the potential of the method on larger models. Additionally, the datasets used, CIFAR-10 and CIFAR-100, are actually quite similar, being both image classification tasks. The main difference between them is the number of classes (10 vs. 100), with CIFAR-100 being an extension of CIFAR-10, adding more categories. However, the feature distributions and task nature are not fundamentally different. Moreover, the results on the extended NEWS dataset are not particularly significant. The comparison methods are also limited (in fact, only one comparison is made). Testing CND on more diverse real-world datasets would make the results more valuable.

**Questions:**

1. Are there any limitations to using CND in models with very deep architectures, where the interpretation of pre-activations might become more complex?

2. Has there been previous research on class-conditional distributions? Could you provide more details on how your work differs from existing research?

---

> ### Author Response · Authors · 2025-11-20
>
> Thank you for your constructive feedback and helpful comments.
>
> ### Weakness 1
>
> We appreciate the reviewer’s observation. The updated manuscript will include clearer descriptions of figures and equations, including a more explicit explanation of the generalization–memorization distinction shown in Figure 2.
>
> ### Weakness 2
>
> We are currently running experiments on Clothing1M and WebVision to demonstrate scalability in real-world settings with larger neural networks.
>
> ### Question 1
>
> At present, we have not observed limitations when applying CND to deeper models, aside from increased computational cost. Very large architectures have not yet been tested and represent a current limitation
>
> ### Question 2
>
> To the best of our knowledge, no prior work studies class-conditional neuron pre-activation distributions, but they are focus only on the marginal distribution.

---

> > ### Comment · Reviewer_skds · 2025-11-28
> >
> > I appreciate the authors' rebuttal, which has addressed most of my concerns. At this stage, I intend to maintain my original score.

---

### Author Response · Authors · 2025-11-20

## General Response

We sincerely thank all reviewers for their constructive and thoughtful feedback. The comments were highly valuable in identifying areas where the paper can be strengthened in clarity, completeness, and experimental depth.

We apologize that the submitted version did not include the appendix. This was due to an uploading error, and it will be included in the next revision.

We are now working on an updated version of the paper that incorporates improvements related to readability, clarity, equations, and figure explanations. A second version will follow later, including additional experiments on larger-scale datasets (Clothing1M and WebVision) with deeper neural networks, as well as further ablations addressing the reviewers’ concerns.

---

> ### Author Response · Authors · 2025-11-23
> **Manuscript Updates**
>
> The manuscript has now been updated in response to the reviewers’ comments:
>
> - The appendix has been added (previously missing).
> - Figures 1 and 2 are now explained in more detail.
> - The equivalence between the marginal and the prior in Equations (4) and (5) has been clarified.
> - Minor text edits and formatting improvements have been applied.
> - A brief overview of the experimental setup has been added in the main text, with full details provided in the appendix.
> - We explicitly clarify that we compute $P(z_j^{(l)} \mid \tilde{y})$ rather than $P(z_j^{(l)} \mid y)$, where $\tilde{y}$ denotes the observed (and possibly corrupted) label. With this clarification, it is now explicit why CND performance degrades under extreme label noise.

---

> ### Author Response · Authors · 2025-12-02
> **Final Version**
>
> We would like to thank all reviewers for their valuable feedback. We believe that the modifications made in response to the major points raised have significantly improved the clarity, rigor, and overall quality of the manuscript.
>
> Below, we summarize the main revisions:
> 	•	The appendix has now been fully included. It contains technical details as well as an analysis of neuron importance for generalization based on the CND.
> 	•	Experiments on WebVision and Tiny-ImageNet have been added. Both use ResNet-50 and demonstrate that the CND can be applied to realistic datasets, further showing its applicability to deeper architectures.
> 	•	A new Appendix F has been added to illustrate how the CND behaves under extreme noise conditions and to explicitly justify why its performance typically degrades in such scenarios.
> 	•	The main CND equation (Eq. 4/5) has been revised to clarify the relationship between the mixture and prior probability distributions.
> 	•	Image descriptions have been improved for clarity.
> 	•	The appendix is now integrated into the main manuscript.
> 	•	Numerous text revisions have been made to enhance readability and consistency.
> 	•	The choice of the levels of noise is justified and coherent across the datasets

---

### Meta-Review · Area_Chair_5tPt · 2025-12-11

**Summary:**

The paper proposed a validation-free early-stopping criterion via class-conditional pre-activation divergence and shows promising small-scale results. However, all reviewers flagged limited empirical scope, namely, evaluations focused on MNIST/CIFAR/NEWS with simple models; larger, realistic datasets and deeper networks were promised, not delivered. Key baselines --- including standard validation-set early stopping and broader recent methods --- were missing. While writing improved in rebuttal, the evidence still remained insufficient to establish robustness and practical utility. Balancing strengths against these persistent gaps, I recommend reject.

**Reviewer Concerns:**

**skds** Concerns: clarity of figures/equations; small models, few datasets/baselines; questions on deeper architectures and prior work. Addressed: clarified figures/equations; promised larger-scale runs. Outstanding: delivered large-scale results and broader datasets/baselines.

**3CiR** Concerns: small datasets; high-noise failure; missing appendix; weak comparisons. Addressed: clarifications, derivation conditions. Outstanding: real large-scale evidence, deeper failure analysis, broader baselines.

**dezB** Concerns: incomplete manuscript, noise-setting justification, missing validation-set early-stopping baseline. Addressed: none. Outstanding: empirical scope/rigor; baseline gap.

**Reviewer Scores:**

I don't think any of the three reviewers would like to increase his or her score.

---

### Decision · Program_Chairs · 2026-01-26

Reject